# Malicious Traffic Identification with Self-Supervised Contrastive Learning

**DOI:** 10.3390/s23167215

**Published:** 2023-08-17

**Authors:** Jin Yang, Xinyun Jiang, Gang Liang, Siyu Li, Zicheng Ma

**Affiliations:** 1School of Cyber Science and Engineering, Sichuan University, Chengdu 610065, China; jinnyangscu@163.com (J.Y.); gangliangscu2022@163.com (G.L.); siyuliscu@163.com (S.L.); zichengmascu@163.com (Z.M.); 2School of Information Science and Technology, Tibet University, Lhasa 850013, China

**Keywords:** network security, malicious traffic identification, transformer, deep learning, long short-term memory (LSTM), self-attention, contrastive learning

## Abstract

As the demand for Internet access increases, malicious traffic on the Internet has soared also. In view of the fact that the existing malicious-traffic-identification methods suffer from low accuracy, this paper proposes a malicious-traffic-identification method based on contrastive learning. The proposed method is able to overcome the shortcomings of traditional methods that rely on labeled samples and is able to learn data feature representations carrying semantic information from unlabeled data, thus improving the model accuracy. In this paper, a new malicious traffic feature extraction model based on a Transformer is proposed. Employing a self-attention mechanism, the proposed feature extraction model can extract the bytes features of malicious traffic by performing calculations on the malicious traffic, thereby realizing the efficient identification of malicious traffic. In addition, a bidirectional GLSTM is introduced to extract the timing features of malicious traffic. The experimental results show that the proposed method is superior to the latest published methods in terms of accuracy and F1 score.

## 1. Introduction

Malicious traffic identification has become a critical approach to detect malicious attacks in information systems. Common types of malicious attacks include Denial of Service (DoS), Distributed Denial of Service (DDoS) attacks, worms, and DNS attacks, among others [1,2]. Particularly, DoS and DDoS attacks can cause extensive damage due to the accessibility and ease of obtaining DoS/DDoS tools, which makes them readily available even to individuals lacking technical expertise [3]. Moreover, as Internet users increasingly prioritize privacy protection and information security, more applications employ encryption mechanisms to secure data transmission. While encrypting data transmission effectively preserves data confidentiality, it also introduces new security risks and threats [4]. Encrypted data can evade security checks from firewalls and intrusion detection systems, enabling network attackers to conceal their attacks and steal data. Detecting encrypted traffic poses greater challenges compared to nonencrypted traffic because the upper layer payload of a data flow becomes invisible after encryption, rendering traditional security systems based on firewalling and deep packet inspection ineffective. Identification methods based on decryption are costly, time consuming, and can raise privacy concerns. Therefore, the accurate identification of malicious traffic under encryption becomes both important and significant [5,6].

In recent years, researchers in the field of malicious traffic identification have shifted their focus to machine learning based on traffic behavior. Traditional machine learning methods, such as support vector machine (SVM), random forest, k-nearest neighbor (KNN), decision tree, etc., can accomplish traffic identification and classification by analyzing the statistical features of malicious traffic [7,8]. However, machine learning methods usually need a manual selection of traffic features, and the establishment of a feature library requires expert knowledge. The quality of feature selection directly affects the classification performance of the model. Therefore, some researchers have introduced deep learning technology into the field of malicious traffic identification so as to eliminate the need for manual feature selection. For example, Wang et al. [9] used a one-dimensional convolutional neural network (1D CNN) to realize the automatic extraction of the spatial features of encrypted traffic. Xing et al. [10] used long short-term memory (LSTM) to extract the time features of network traffic. Lin et al. [11] combined a convolutional neural network (CNN) with a recurrent neural network (RNN) and built a model called TSCRNN. While these methods can to some extent identify malicious traffic, they still suffer from low accuracy, particularly when distinguishing encrypted malicious traffic. Additionally, these existing traffic identification methods can only be trained by using labeled traffic data, which limits their ability to learn general features from unlabeled data. This limitation becomes evident when the captured malicious traffic lacks labels or when there is a scarcity of labeled data, leading to algorithmic convergence issues and ineffectiveness.

This paper proposes a novel model for malicious traffic identification to address the aforementioned issues. The proposed approach employs self-supervised contrastive learning to pretrain the model, allowing it to learn common semantic information from unlabeled malicious traffic data. This knowledge is then transferred to downstream tasks, leading to faster convergence in these tasks. Furthermore, the approach utilizes both a Transformer and GLSTM (GELU-LSTM) as the backbone networks for feature extraction. These networks capture the byte-level and temporal features of malicious traffic, enhancing the representation of malicious patterns and significantly improving the algorithm’s identification accuracy. The main contributions of our paper can be summarized as follows:A malicious-traffic-identification method based on contrastive learning is proposed. Our method has shown superiority for malicious traffic identification compared with traditional methods relying on labeled samples, which can process arbitrary unlabeled packet capture files into vectorized traffic representations and learn data feature representations carrying semantic information from unlabeled data, thus improving the model accuracy.The proposed model employs a self-attention mechanism to accurately extract bytes features of malicious traffic. Compared with the convolutional-neural-network-based feature extraction module, the Transformer-based feature extraction module can significantly improve the feature extraction capability by capturing key features of the malicious traffic as well as learning the correlation between multiple features.A bidirectional GLSTM (bi-GLSTM) is proposed to extract the temporal features of malicious traffic, which uses the GELU nonlinear function as the activation function in the recurrent stage. The idea of stochastic regularization is introduced in the activation process to enhance the generalization ability of the model, which makes bi-GLSTM more suitable for processing traffic data than the conventional bi-LSTM network.

The rest of the paper is structured as follows. The related work is summarized in Section 2, and the framework of the algorithm and each module are introduced in detail in Section 3. The performance of the proposed method is evaluated in Section 4 and compared with other state-of-the-art methods. Finally, the limitations of the proposed method are discussed in Section 5, and the paper is concluded in Section 6.

## 2. Related Work

In recent years, the research on malicious traffic identification has undergone a gradual shift from obtaining matching rules through manual operation based on experience to AI-assisted adaptive feature extraction, from using a single identification algorithm to combining multiple algorithms, and from a single-point traffic analysis and judging to distributed network identification.

Among the recently published studies, Wang et al. [6] systematically described the often-used methods of detecting encrypted traffic and used the feature-based machine learning method to detect network traffic. In Reference [12], Jaber et al. proposed a network application traffic-detection method that analyzes both the statistical information of traffic and the communication information of the host. In Reference [13], Zhang et al. proposed a maliciously encrypted traffic-identification algorithm combining supervised and unsupervised learning. This method realizes the classification of data flows based on the statistical flow characteristics and packet payload characteristics. Cheng et al. [14] employed the Isolation Forest (iForest) to detect malicious traffic in the network. iForest demonstrates faster convergence when dealing with smaller datasets. In Reference [15], Xiong et al. utilized the One-class Support Vector Machine (OC-SVM) to detect abnormal traffic in the network. Paulauskas et al. [16] applied the Local Outlier Factor algorithm (LOF) to detect anomalies or malicious flows in large data streams. This method relies on manually selected feature sets. Draper Gil et al. [17] studied the time-dependent characteristics of VPN traffic and used the decision tree algorithm, a typical machine learning algorithm, to group VPN data flows into different categories according to the data flow type. Vincent et al. [18] proposed a method called AppScanner, which uses deep learning technology to identify mobile device applications from malicious network traffic. Thakkar et al. [19] proposed an ensemble learning approach that combines Deep Neural Networks (DNNs) with Bagging classifiers to detect network attack behavior. Lotfollahi et al. [20] proposed an encrypted traffic-classification framework called deep packet, which can not only identify encrypted traffic but also distinguish VPN and non-VPN data flows. Wang et al. [9] introduced a convolutional neural network (CNN) into traffic classification and developed a traffic classification method that integrates feature extraction and classification modules. Xing et al. [10] used an LSTM-based automatic encoder to detect maliciously encrypted data flows. Zhang et al. [21] used a 3D-CNN network to enable the automatic classification of encrypted sessions. Ibitoye et al. [22] introduced a novel Self-normalizing Neural Network (SNN) to achieve attack traffic classification. Additionally, the authors discussed the performance of Feed-forward Neural Networks (FNNs) in classifying network attack traffic. References [11,23] propose a parallel model combining RNNs and CNNs to classify the flow of application. Shen et al. [24] combined the KNN, random forest, and decision tree algorithms to realize the classification of encrypted web page data based on the features extracted from the two-way interactions between client and server. Alghanam et al. [25] proposed an ensemble learning approach based on LS-PIO (Local Search with a Pigeon-Inspired Optimizer) by integrating multiple machine learning algorithms and enhancing the feature extraction algorithm, the Pigeon-Inspired Optimizer (PIO). Reference [26] proposes an LSTM network for classifying VPN data flows based on an attention mechanism.

## 3. Proposed Model

This section will introduce the proposed malicious-traffic-identification model in detail. The model is illustrated in Figure 1. The malicious traffic identification in this paper focuses on the multiclassification problem, aiming to classify malicious traffic into specific classes (normal traffic, Distributed Denial of Service, Data Theft, etc.). The proposed model is divided into three main phases: data preprocessing, pretraining, and transfer learning. During the data preprocessing phase, arbitrary untagged packet capture files are processed into vectorized traffic representations. The pretraining stage involves training the model by using self-supervised contrastive learning on the unlabeled dataset. Finally, in the transfer learning phase, the pretrained model is fine tuned on the labeled data to adapt to the new scenario.

To tackle the problem of lacking the ability to extract global information from malicious traffic, which is common with the existing methods, we use a Transformer as the backbone network of the malicious-traffic-identification model to extract features from the malicious traffic. The network adopts a self-attention mechanism based on the shifted window, which brings two benefits: Firstly, the application of the self-attention mechanism in the global calculation of the input data overcomes the limitation of previous algorithms in extracting context information from bytes, thereby significantly improving the accuracy of detecting and classifying malicious data flows. Secondly, by limiting the operation to the window, the amount of operation is greatly reduced, making it more suitable for processing flow data.

### 3.1. Overview

In this work, we preprocess the raw traffic data (i.e., packet capture files) into two-dimensional data matrices that the model can directly process. These 2D data matrices contain various traffic features, such as malicious traffic byte information and packet temporal information. These features serve as discriminative characteristics to differentiate between different types of traffic. Since the byte data in network traffic transmission carries contextual semantics, it can also be considered as semantic information.

The proposed model takes 2D data matrices as the input and adaptively extracts the traffic features through the Transformer module and GLSTM module, ultimately obtaining the traffic’s embedding representation in a high-dimensional space. In the pretraining phase based on contrastive learning, unlabeled traffic data are used to train the model, allowing it to learn general semantic information and temporal features from unlabeled traffic data. During this phase, the high-dimensional embedding representation is further mapped onto the unit sphere, aiming to maximize the distance between the representations of positive and negative samples on the unit sphere by reducing the loss value of the contrastive loss function in Section 3.3.4. Meanwhile, the representations of positive samples are encouraged to be closer to each other on the unit sphere. In the transfer learning phase, the pretrained model parameters are utilized to expedite the model convergence. Additionally, a new fully connected layer is added to match the downstream task, typically composed of a linear layer. In the fine-tuning stage of transfer learning, the representations of the traffic samples of the same category in the high-dimensional space should be as close as possible while the representations of the traffic samples from different categories should be as far apart as possible. In the testing phase of transfer learning, the samples that are not involved in training are used to comprehensively evaluate the model’s performance.

### 3.2. Data Preprocessing

During data preprocessing, the raw traffic data (i.e., the packet capture files) are processed into a format that can be directly input into the model for inference.

The raw data files are divided according to communication sessions; the packets with the same five tuple (source/destination IP, source/destination port, and protocol) belong to the same session stream, and the packets that may be retransmitted are discarded. In order to avoid the impact of the frequently changing IP addresses and port numbers in the encrypted traffic on the recognition accuracy of the model, we randomize the IP addresses and port numbers in the session during the data preprocessing stage; specifically, we remove the bytes where the IP addresses and port numbers are located in the packets. For each session, the packets are arranged in timestamp order, and only the first N packets in the session are retained. The first L bytes of each packet are reserved to represent the packet. Sessions with more than N packets are truncated, while sessions with fewer than N packets are padded with zeros. The processing of packets follows the same principle. The processing results in a sequence of n malicious traffic of length N*L, with n denoting the total number of sessions.

In order to preserve the information carried by the malicious traffic as much as possible and to make it suitable for machine processing, each byte of the malicious traffic sequence is converted to a decimal value within the range [0, 255]. In order to make the traffic features in the one-dimensional space easier to extract and exploit, we reshape the one-dimensional traffic sequence into a two-dimensional traffic matrix to represent the traffic. Specifically, the two-dimensional traffic matrix maintains the same length and width and a depth of value one. Since N × L is not guaranteed to be a squared value, the length and width of the two-dimensional matrix can be scaled with an upper bound of N × L. In this work, we set N = 7, L = 125, and the length and width of the two-dimensional matrix to 28 × 28.

### 3.3. Pretraining with Self-Supervised Contrastive Learning

Our proposed pretraining task with self-supervised contrastive learning [27] aims to make the model learn the intrinsic semantic information of the traffic samples by constructing a contrastive task and characterizes the manifold structure of the sample space by constructing an augmented view for each sample. The detailed flowchart is shown in Figure 1.

#### 3.3.1. Contrastive Task Construction

The construction of the contrastive task is a crucial step in contrastive learning. In this step, the samples are grouped according to their intrinsic semantics. The samples with similar semantics are grouped into positive pairs, while the samples with different semantics form negative pairs. Furthermore, to capture the semantic relationship information between the samples, it is necessary to construct augmented samples for each sample. Current research on augmented samples mainly focuses on image data, where methods such as scaling, rotating, cropping, and adding Gaussian noise are used to generate sample pairs of the image data. However, converting traffic data into images and then constructing image-enhanced sample pairs can be extremely time consuming. To avoid such high resource-consuming computations when constructing samples with similar semantic relationships to the original traffic samples, we construct the enhanced samples by generating random masks.

The construction of sample pairs is illustrated in Figure 2. In this process, each value in the input matrix is randomly masked with a probability of *k*. The mask matrix is a two-dimensional matrix of the same size as the input matrix, containing *k* zeros at random positions. The generation of masked samples can be represented by the following equation:(1)$$session_{masked}=session\times mask$$

#### 3.3.2. Contrastive Encoder

A deep feature encoder is employed to conduct feature extraction on the traffic data representation. The proposed feature extraction module is divided into two parts: one part utilizes the Transformer to extract the packet byte features while the other part utilizes long short-term memory networks to capture temporal relationships. Given a sample of input network traffic, denoted as *x*, the encoder can be written as follows:(2)x′=Enc(x)∈ ℝb where x ∈ X
where *Enc*(.) represents the encoder function. In this paper, the encoder is composed of the Transformer module and GLSTM module. The details of these modules are presented in Section 3.4 and Section 3.5 respectively.

#### 3.3.3. Contrastive Projector

The projector module maps the extracted network features onto the unit sphere and is used to calculate the similarity score of the samples. The feature vectors corresponding to similar samples should have a smaller distance on the unit sphere, while the samples with larger differences correspond to feature vectors that are farther apart on the unit sphere. The projector module normalizes the traffic features to form a standardized vector representation and subsequently feeds this representation through a fully connected layer to derive the embedded feature representation. Lastly, the output is normalized to the unit sphere. After mapping the features onto the unit sphere, we obtain the normalized feature vector with its unit size. Here, we use the cosine similarity to measure the similarity between two vectors. It measures the cosine of the angle between two vectors in a high-dimensional space. If the angle between the two vectors is 0 degrees, the cosine similarity is one. Otherwise, it outputs a number less than 1 with a minimum value of −1. Specifically, we use the dot product to compute the cosine similarity. Given the input feature x, the projector layer can be represented as follows:(3)r=Map(x)=Map(Enc(x))
where *Map*(.) represents the projector function. In this paper, the projector is constructed by using two linear layers alternating with a ReLU activation function layer.

#### 3.3.4. Contrastive Cross-Entropy Loss

The loss function is the core of contrastive learning. Given a sample *x* as an anchor point, *x*+ is the enhanced view from the same original image as *x*, and *x*− is the view from a different original image than *x*. The contrastive loss function is constructed to ensure that similar sample pairs (*x*, *x*+) are close to each other while dissimilar sample pairs (*x*, *x*−) are far from each other. The contrastive loss function is defined as:(4)$$Loss=\sum_{i}\frac{-1}{P(i)}\sum_{p\in P\left(i\right)}log\left(\frac{exp(r_{i}\;\cdot \;r_{p})}{\sum_{a\in A\left(i\right)}exp(r_{a}\;\cdot \;r_{p})\;}\right)$$
where *P*(*i*) and *A*(*i*) denote the positive pair set and the complete pair set of the anchor ***r****_i_*, respectively.

The contrastive loss function minimizes the embedding distance of the positive sample pairs (*x*, *x*+) and maximizes the embedding distance of the negative sample pairs (*x*, *x*−) by comparing the similarity of the positive and negative sample pairs. In this way, the network model gradually learns the specific coding ability to transform similar inputs into feature vectors that are close in distance in a high-dimensional space, and the greater the difference, the greater the distance between the feature vectors. With such capabilities, the network can effectively perform downstream tasks, exhibiting rapid convergence and superior performance compared to learning from the random initialization of states. As the network has learned how to distinguish different data based on the data structure and has specific knowledge based on the traffic data, it is better equipped to handle downstream-related tasks.

### 3.4. Transformer Module

In the Transformer module, modeling and calculations are performed on the traffic feature map obtained in the preprocessing stage to obtain the bytes features of the traffic. Then, the obtained features are evaluated automatically, and the features with a strong correlation will be evaluated at first. Because the model uses the window-based multihead self-attention (W-MSA) mechanism and confines the operations within the window, the context dependencies can be obtained with much less model training time.

As shown in Figure 3, the Transformer module consists of four stages. Except for stage 1, each stage consists of several consecutive Transformer blocks and patch-merging modules. In stage 1, the feature image is first divided into contiguous patches through patch partition, and the patches are input into the module in parallel. Then, the high-dimensional data are mapped to the low-dimensional space by the linear embedding layer. Finally, the obtained data are input into the Transformer blocks for feature extraction. In the proposed model, the Transformer blocks are always used in pairs. The structure of one single Transformer block in the Transformer model is shown in Figure 4.

The Transformer block comprises multiple computing units such as W-MSA, multilayer perceptron (MLP), and the normalization layer (LN), which are connected through the residual structure. Compared with the original multihead self-attention (MSA) mechanism [28], the most prominent feature of W-MSA is the window partition. W-MSA divides the original input image into multiple windows and applies MSA in each window for calculations, thus changing the original larger operation window into smaller operation windows. This arrangement greatly reduces the operation workload and improves efficiency, resulting in a higher operation speed and reduced training time. The formula for calculating the MSA is as follows:(5)$$MAS=Concat\left(head_{1},\ldots ,head_{h}\right)W^{O}$$
where $head_{i}=Attention\left(QW_{i}^{Q},KW_{i}^{K},VW_{i}^{V}\right)$, $Attention\left(Q,K,V\right)=soft\max\left(\frac{QK^{T}}{\sqrt{d}}\right)V$.

The computing unit of the second Transformer block changes from W-MSA to SW-MSA (shifted window-based multihead self-attention). As mentioned earlier, although the W-MSA module divides the input features according to the windows, thus reducing the workload of the calculation, there is a lack of an information interaction between the windows, making it impossible to realize a global operation. The SW-MSA module is used to solve this particular problem. Compared with the W-MSA module, the SW-MSA module has an extra function of window shifting. Before performing a window operation on the input feature data, the model needs to carry out a shifting operation on the feature tensor. Consequently, some previously nonadjacent data blocks may be grouped into the same operation window for window operation, and this is the information interaction between different windows. The main function of the patch-merging module is to downsample the input data. After the treatment of the patch-merging module, the length and width of the input feature data unit are reduced to half of the original values, and the depth is doubled. The operation formula of the Transformer block is shown in Equations (6)–(9):(6)$${\hat{x}}^{l}=W\text{-}MSA(LN(x^{\left(l-1\right)}))+x^{\left(l-1\right)}$$
(7)$$x^{l}=MLP(LN({\hat{x}}^{l}))+x^{l}$$
(8)$${\hat{x}}^{\left(l+1\right)}=SW\text{-}MSA(LN(x^{l}))+x^{l}$$
(9)$$x^{\left(l+1\right)}=MLP(LN({\hat{x}}^{\left(l+1\right)}))+{\hat{x}}^{\left(l+1\right)}$$

Next, we will describe the difference in computational complexity between the MSA module and the W-MSA module. For a patch with a size of h×w, Equation (10) represents the computational complexity when the MSA mechanism is directly used, and Equation (11) represents the computational complexity when the W-MSA mechanism is used:(10)$$\Omega \left(\mathrm{MSA}\right)=4hwC^{2}+2{\left(hw\right)}^{2}C$$
(11)$$\Omega \left(W\text{-}MSA\right)=4hwC^{2}+2M^{2}hwC$$
where *C* represents the dimension of the data in the depth direction after the data pass through the linear embedding layer and *M* represents the window size. It can be seen that the computational complexity when the MSA mechanism is directly used is the square of hw, while W-MSA is linear when M is fixed. This further highlights the advantage of the Transformer in the model calculations.

### 3.5. Bidirectional GLSTM Module

For the network traffic, regarding the data with a temporal relationship, it is important to be able to consider the temporal relationship during feature extraction. Therefore, in this work, a feature extraction is performed by using a long- and short-term memory network instead of a convolutional neural network.

At present, the most widely used LSTM (RNN-based structure) [29] uses tanh as the activation function of the recurrent step. However, the problem of feature information renormalization arising from using tanh will lead to the loss of some important time features. To solve this problem, this paper proposes a new LSTM network structure: GLSTM (GELU-LSTM), which uses the GELU (Gaussian error linear unit) [30] as the nonlinear function of the recurrent step, as shown in Equation (12). And, the unit structure of GLSTM is shown in Figure 5. The idea of stochastic regularization is introduced in the activation process of the GELU, which enhances the probability interpretation for the activation function and renders the model more robust. Compared with tanh, the activation function GELU can better solve the gradient disappearance problem of the RNN and retain more timing information of the input data in the recurrent stage, so it is more suitable for processing the traffic data.
Figure 5The cell structure diagram of the proposed GLSTM.
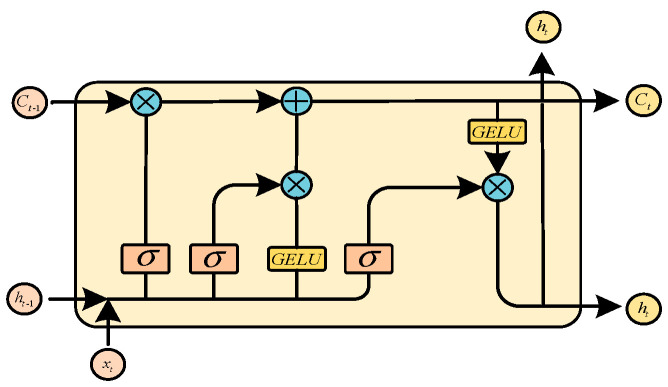

(12)GELU(x)=xP(X≤x),x~N(0,1)

The GLSTM network uses three gating units and one storage unit to manage the information flow. The three gating units are the input, output, and forgetting gate. The storage unit is used to save historical information. GLSTM uses the forgetting gate to retain or discard the historical information in the storage unit. The formula is as follows:(13)$$f_{t}=\sigma (W_{fx}x_{t}+W_{fh}h_{t-1}+b_{f})$$
wherein $x_{t}$ represents the input information at the time t, $W_{fx}$ represents the weight matrix of the input information, $h_{t-1}$ represents the output state of the hidden layer at the previous moment, $W_{fh}$ represents the weight matrix of the hidden layer, and $b_{f}$ represents the offset information. Similarly, the calculation formulas of the input gate and output gate are as follows:(14)$$i_{t}=\sigma (W_{ix}x_{t}+W_{ih}h_{t-1}+b_{i})$$
(15)$$o_{t}=\sigma (W_{ox}x_{t}+W_{oh}h_{t-1}+b_{o})$$

The candidate state $g_{t}$ and updating state $c_{t}$ determine which pieces of effective information can be retained for the next neuron in the information flow stage. The formula is as follows:(16)$$g_{t}=GELU(W_{gx}x_{t}+W_{gh}h_{t-1}+b_{g})$$
(17)$$c_{t}=f_{t}⊙c_{t-1}+i_{t}⊙g_{t}$$

Finally, the updating state $c_{t}$ is made to undergo an XNOR operation with the output gate $o_{t}$ through a GELU activation function to obtain the output of the current GLSTM cell unit, as shown in Equation (18):(18)$$h_{t}=GELU(c_{t})⊙o_{t}$$

Bidirectional GLSTM includes not only the state of the forwarding operation but also the state of the backward propagation. The output features obtained through the Transformer module will be input into the GLSTM cell unit in two directions, as shown in Equation (19):(19)$$h_{t}=\vec{h_{t}}⊕\overset{\leftarrow }{h_{t}}$$
where → represents the forward propagation state and ← represents the backward propagation state. The output states in both directions are obtained through the XOR operation. In summary, the proposed bi-GLSTM module structure is shown in Figure 6.

## 4. Experiment and Analysis

### 4.1. Dataset

To verify the feasibility of the proposed method, we tested it on the Bot-IoT dataset [31,32,33], which was created by Cyber Range Lab of UNSW Canberra in 2018. The Bot-IoT dataset contains data of normal traffic and four malicious traffic types, including the DDoS, DoS, Scan, and Data Theft. Since the total number of sessions in the Bot-IoT dataset exceeded 72 million, we used 2% of the original dataset for the experiment, and the sample size statistics of the dataset used are shown in Table 1. Each session contained in the dataset is generated by IoT devices widely used in the IoT network, making the dataset a favorable choice for most researchers.

The total number of sessions obtained through segmentation is 1,143,786. The category name of each type of malicious traffic is input into the proposed model as label data. The experimental dataset was divided into an unlabeled set (40%), a training set (40%), and a testing set (20%). Each category was further divided into three sets according to the same proportions. The samples were randomly allocated so as to ensure the objectivity of the verification. The hardware platform and experimental parameters used in the experiment are shown in Table 2 and Table 3, respectively.

### 4.2. Experimental Setup

In this paper, four classical evaluation indicators, namely accuracy (ACC), precision (PR), the recall rate (RC), and the false positive rate (FPR), as well as the F score, are used to compare the performance of the proposed method with that of other methods. The formulas for calculating the ACC, PR, RC, and FPR are shown in Equations (20)–(23):(20)$$ACC=\frac{TP+TN}{TP+TN+FP+FN}$$
(21)$$PR=\frac{TP}{TP+FP}$$
(22)$$RC=\frac{TP}{TP+FN}$$
(23)$$FPR=\frac{FP}{FP+TN}$$

Precision and the recall rate are generally mutually constraining. The F score reflects both the precision and recall. β is used to adjust the proportions of precision and the recall rate, as shown in Equation (24). One often-used metric is the F1 score (β = 1), which assigns equal importance to precision and the recall rate. The formula for calculating the *F*1 score is shown in Equation (25):(24)$$F\text{-}S\mathrm{core}=(1+\beta^{2})*\frac{PR*RC}{\beta^{2}*PR+RC}$$
(25)$$F1\text{-}Score=2\;*\;\frac{PR*RC}{PR+RC}$$

The average value of each metric can reflect the real performance of the model. It is defined in Equation (26), where *N* represents the total number of categories and *X* represents a certain evaluation metric. For example, when *X* is precision, the value obtained by using the formula is the macro average of precision:(26)$$Macro-average=\frac{\sum_{i=1}^{N}X_{i}}{N}$$

### 4.3. Experimental Results

To evaluate the performance of the proposed algorithm in terms of its identification accuracy, we carried out experiments to make a comparison between the proposed algorithm and seven other typical malicious-traffic-identification algorithms, namely iForest [14], the OC-SVM [15], LOF [16], DNNs [19], FNNs [22], SNNs [22], and LS-PIO-based ensemble [25]. These methods were introduced in Section 2. According to the classification of traditional machine learning and deep learning, the traditional machine learning methods include iForest [14], the OC-SVM [15], LOF [16], and LS-PIO-based ensemble [25]. The deep learning methods comprise DNNs [19], FNNs [22], SNNs [22], and the proposed method.

Figure 7 shows the accuracy and Macro-F1 scores of different algorithms. From the results, we know that the malicious-traffic-identification method based on deep learning has a higher accuracy than the method based on traditional machine learning. This is because the deep-learning-based method can adaptively extract traffic features without manual participation in feature extraction, which can more fully extract the features of the original traffic. Moreover, deep-learning-based methods usually use raw traffic data as the input instead of manually extracted traffic features. As a result, the model can learn more abstract and powerful representations from the raw data, leading to a more accurate classification. For example, the OC-SVM [15], which is based on traditional machine learning algorithms, produces a 77.79% ACC and 76.28% Macro-F1 score, whereas the SNN [22], which is based on deep learning, produces a 91% ACC and 91% Macro-F1, which is significantly better than the OC-SVM [15]. By integrating multiple traditional machine learning methods, [25] obtained a 98.73% ACC and 96.70% Macro-F1, which was the best result of all the traditional machine learning algorithms. However, its performance is easily surpassed by the deep-learning-based DNN [19], where the DNN achieves a 98.99% ACC. Through the malicious-traffic-identification method based on deep learning proposed in this paper, the identification ACC can be further improved to 99.48%, and the Macro-F1 can be increased to 99.46%. The proposed method outperforms the OC-SVM [15] by 21.69% and outperforms the SNN [22] by 8.48% in terms of the ACC. It is 23.18% higher than the OC-SVM [19], 8.46% higher than the SNN [22], and 4.51% higher than the DNN [19] in terms of the Macro-F1. Overall, the proposed method is the best performing of the experimental methods and represents a significant improvement over the existing methods.

In order to illustrate the effectiveness of the pretraining method, Table 4 presents a comparison between the nonpretrained approach and the proposed method under the same experimental parameters. From Table 4, it can be observed that the method using the pretraining method achieves a higher ACC and demonstrates a superior performance in terms of the Macro-F1, Macro-PR, Macro-RC, and Macro-FPR. In addition, to show the overall identification performance, we also demonstrated a more fine-grained identification result. Table 5 shows the identification results of the precision, recall, F1, and FPR obtained by the proposed method for the identification of each subcategory. From Table 5, it can be seen that the proposed method has a high recognition accuracy and very low false recognition rate even for the more fine-grained malicious traffic categories. In order to test the influence of different loss functions on the experimental results, we used different loss functions to train the model, and the results are shown in Table 6.

We further analyzed the reasons behind the high performance of the proposed algorithm in multicategory identification tasks. Firstly, the proposed model segments the original data according to the session; thus, the timing relationship of the data packets and the interaction information between the two applications are preserved to the greatest extent. Secondly, the proposed model only uses the first several bytes of each session because the data packets at the beginning of the session contain data that can be used to distinguish the traffic types. Thirdly, the proposed model adopts a self-attention mechanism with a strong generalization ability and can perform a better feature extraction on the input data. Compared with traditional CNN-based feature extraction algorithms, the proposed model has a larger receptive field. Therefore, it can effectively capture long-distance dependent features. In addition, the use of LSTM makes it more suitable for processing time-dependent traffic data. Finally, the use of pretraining based on self-supervised contrast learning allows the model to fully learn the feature representation of the data carrying semantic information, thus benefiting the downstream task.

For a more intuitive demonstration of the identification performance of the proposed SW-GLSTM model, the confusion matrices generated by the proposed method are visualized and shown in Figure 8.

## 5. Discussion

In this paper, we propose a self-supervised contrastive learning-based method for identifying malicious traffic, which allows for the utilization of unlabeled malicious traffic for training. By mining the common features from the unlabeled data, the method transfers this knowledge to downstream tasks. The experimental results demonstrate the accuracy of the proposed algorithm in identifying malicious traffic. However, the method still has certain limitations. Due to time constraints, the current data scale used in the experiments is limited. In future work, we aim to expand the data scale to real-world scenarios, thereby enhancing the model’s ability to recognize large-scale data and improving its generalization capabilities.

## 6. Conclusions

This study introduces a novel malicious-traffic-identification method based on self-supervised contrastive learning. The method enables the transformation of any unlabeled packet capture files into vectorized traffic representations and employs contrastive learning to extract common semantic information about malicious traffic from the unlabeled data, which is then transferred to downstream tasks. The proposed method consists of three main stages: data preprocessing, pretraining, and transfer learning. In the data preprocessing stage, the raw traffic data are initially processed to obtain 2D data matrices as the input. During the pretraining stage, contrastive learning is employed to extract generic features of the malicious traffic from the unlabeled data and transfer them to downstream tasks. In this stage, the input data undergo feature extraction by using the Transformer module to capture byte-level features, followed by the GLSTM module to extract temporal features, resulting in high-dimensional vectors representing the network traffic after feature extraction. In our approach, we consider the Transformer and GLSTM as an integrated unit to extract both byte-level and temporal features of the network traffic, which serve as discriminative characteristics for traffic identification. In the transfer learning stage, the pretrained model is fine tuned by using labeled data, and then its performance is evaluated by using unlabeled data. The experimental results on the real-world malicious traffic datasets demonstrate that the proposed method achieves the highest performance in malicious traffic identification, with an accuracy of 99.48% and a Macro-F1 score of 99.46%, outperforming state-of-the-art methods.

## Figures and Tables

**Figure 1 sensors-23-07215-f001:**
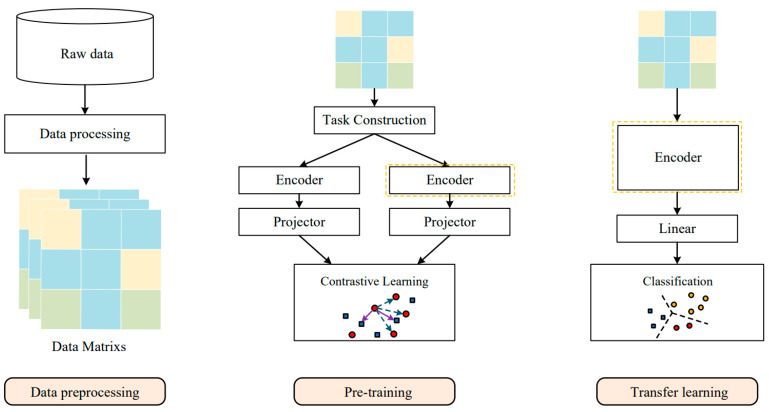
Workflow of the proposed method. After pretraining, the encoder will be used for transfer learning for downstream tasks.

**Figure 2 sensors-23-07215-f002:**
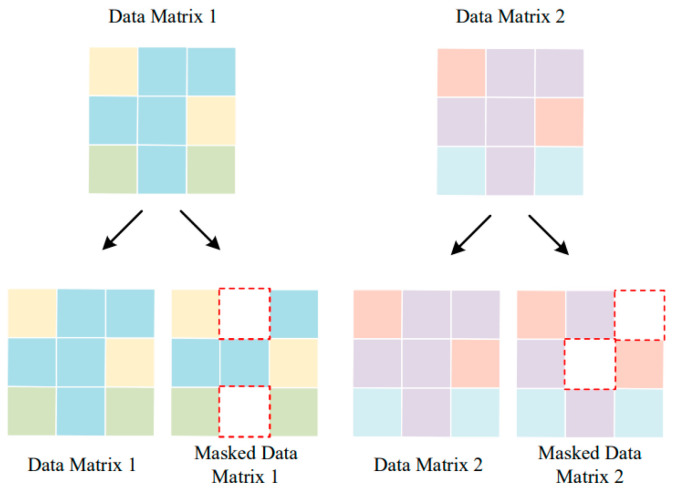
Sample pair construction based on random masking, with number of masked positions *k* = 2. Data Matrix 1 forms a positive pair with Masked Data Matrix 1, and Data Matrix 1 forms a negative pair with Data Matrix 2 (or the Masked Data Matrix 2).

**Figure 3 sensors-23-07215-f003:**
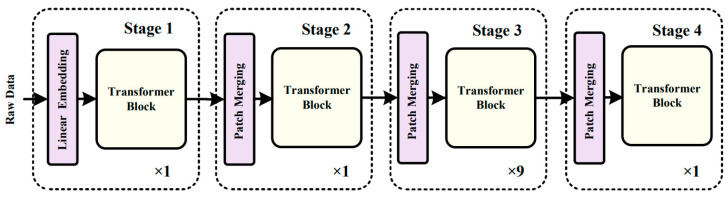
The structure of the Transformer model.

**Figure 4 sensors-23-07215-f004:**
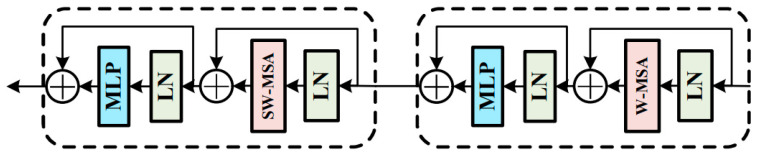
The structure of one single Transformer block in Transformer model.

**Figure 6 sensors-23-07215-f006:**
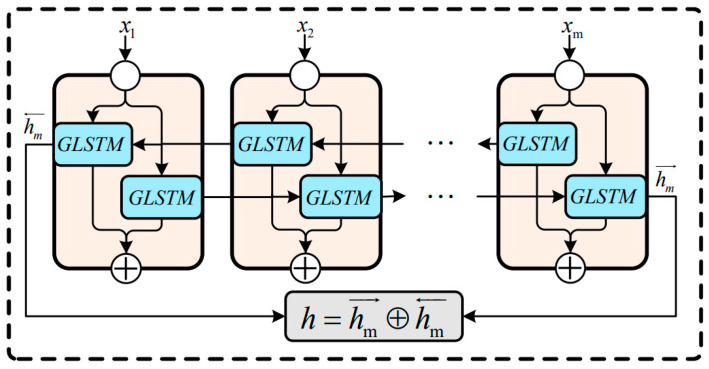
The structure of the proposed GLSTM.

**Figure 7 sensors-23-07215-f007:**
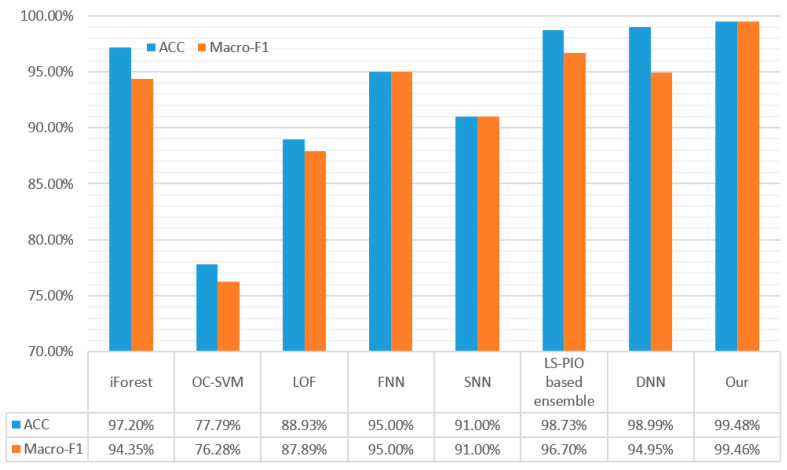
Classification ACC and Macro-F1 score of different algorithms.

**Figure 8 sensors-23-07215-f008:**
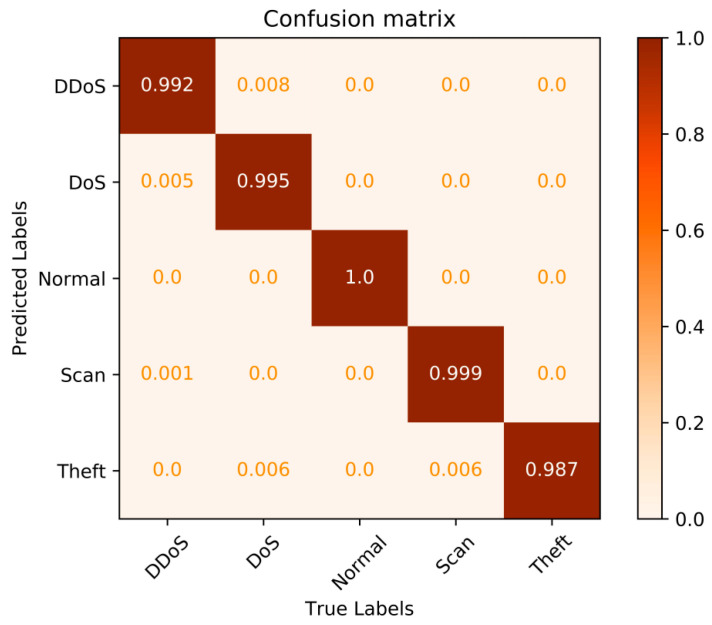
Confusion matrices of the proposed model.

**Table 1 sensors-23-07215-t001:** Description of the Bot-IoT dataset.

Classes	Number
DDoS	534,364
DoS	535,358
Normal	840
Scan	72,442
Theft	782
Total	1,143,786

**Table 2 sensors-23-07215-t002:** Experiment platform specification.

Item	Specifications
Op. Sys.	Ubuntu 16.04.6 LTS
Python	3.8.7
Pytorch	1.12.1+cu116
GPU	2× NVIDIA GeForce RTX 3080 12 GB
RAM	64 GB DDR4 @2666 MHz
Nvidia Driver	531.79
CUDA Driver	11.6

**Table 3 sensors-23-07215-t003:** Hyperparameters used in the experiments.

Item	Hyperparameters	Item	Hyperparameters
Optimizer	Adam	Sequence Length of GLSTM	1
Loss Function	Cross-Entropy	Dropout Ratio	0.25
Learning Rate	0.0001	Activation Function	ReLU
Batch Size	256	Input Dimension	28 × 28 ^T^, 1024 ^G^
Epoch	1000 *, 30 ^+^	Output Dimension	1024 ^T^, 64 ^G^
Number of Views	2	Embedding Dimension	128 ^T^, 256 ^G^
Temperature Coefficient	0.07	Layers	4 ^T^, 2 ^G^

*: pretraining step, ^+^: transfer learning step; ^T^: transformer module, ^G^: GLSTM module.

**Table 4 sensors-23-07215-t004:** Experimental results with and without pretraining.

Methods	ACC	Macro-PR	Macro-RC	Macro-F1	Macro-FPR
Without pretraining	97.97%	96.63%	85.09%	88.68%	0.60%
With pretraining (ours)	99.48%	99.45%	99.47%	99.46%	0.16%

**Table 5 sensors-23-07215-t005:** The identification result of proposed method on Bot-IoT dataset.

Classes	PR	RC	F1	FPR
DDoS	99.20%	99.40%	99.30%	0.45%
DoS	99.45%	99.20%	99.32%	0.32%
Normal	100%	99.40%	99.70%	0.00%
Scan	99.89%	99.98%	99.93%	0.04%
Theft	98.73%	99.36%	99.04%	0.01%

**Table 6 sensors-23-07215-t006:** The results of using different loss functions.

Loss Function	ACC	Macro-PR	Macro-RC	Macro-F1
Cross-Entropy	99.48%	99.45%	99.47%	99.46%
NLL Loss	98.15%	78.80%	76.66%	77.62%
MultiFocal Loss	98.39%	98.52%	97.22%	97.85%

## Data Availability

Data associated with this research can be retrieved online from the Cyber Range Lab of UNSW Canberra via https://research.unsw.edu.au/projects/bot-iot-dataset (accessed on 10 August 2023).

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
