# Peer review of "Malicious Traffic Identification with Self-Supervised Contrastive Learning"

_sensors, 2023, doi:10.3390/s23167215_

Round 1

Reviewer 1 Report

The manuscript presents a study for Malicious Traffic Identification with Self-Supervised Contrastive Learning. The idea is perfect, but the manuscript can be improved by considering the following comments.

  1. I suggest the authors add citations to the first paragraph of the introduction section.
  2. The research gap (problem) should be described clearly in the introduction section.
  3. Add a description of the manuscript’s structure at the end of the introduction section.
  4. Add a description for the variables in equations in the manuscript.
  5. The main caption of Figure 3 is missing.
  6. I suggest the authors include a short introduction about the algorithms used for comparison against the proposed model in terms of accuracy as mentioned in Figure 6.
  7. What are the main features used for this study and how can be extracted? And how did the authors perform data preprocessing before using the datasets for training the proposed model? Please mention this information in the manuscript.
  8. The authors can add more details about the hardware and software used for conducting the experiment.
  9. As the authors used deep learning methods in their study, it is better to provide a loss of function comparison to show how the training process works.  
  10. I suggest the authors add a discussion section to discuss the meaning of the study’s results and the limitations of the study.
  11. The content of the conclusions section is written simply. Please rewrite it considering a description of the methodology, the significance of the study and results, and future work as well.
  12. English grammar errors and typos should be checked and modified; the whole manuscript should be checked carefully.

Reviewer 2 Report

1. could authors talk about transferibility of the proposed model ?

2. could authors talk about efficiency of the proposed model ?

3. could authors talk about novelty of the manuscript ?

Some typo can be modified.

Reviewer 3 Report

This paper try to propose a malicious traffic identification method based on contrastive learning. This paper uses Transformer to extract bytes features and GLSTM to extract the timing features of malicious traffic. However, the writing and evaluation work are Weak. To become a research paper, some efforts need to be further done.

1. The new contribution is not obvious. Section 2 list many papers, but the comparisons are not given. The proposed method is compared with [23-28] in sub-section 4.3. Also, the differences are not given. In fig.1, is transfer learning the transformer module?

2. In section 4, the evaluation work does not reflect the proposed model. The evaluation parameters are not given. From the evaluation, it cannot see the procedure of extract the bytes features and the timing features of malicious traffic. How to verify the proposed model is high efficiency.

Extensive editing of English language required. It is difficult to understand the methods and results.

Round 2

Reviewer 3 Report

I am glad that authors revised this paper based on comments. This paper can be accepted after major revision. One point is that the in section 4.1, the experimental dataset is Bot-IoT dataset 400 [20]-[22], and was divided into an unlabeled set (40%), a training set 410 (40%) and a testing set (20%). In section 3, the pre-training stage involves training the model using self-supervised contrastive learning on the unlabeled dataset. The reviewer does not see the proposed pre-training method is evaluated.

Moderate editing of English language required.

Round 3

Reviewer 3 Report

I am glad that authors revised this paper based on comments. This paper can be accepted.

 Minor editing of English language required。